# Read Count Patterns and Detection of Cancerous Copy Number Alterations in Plasma Cell-Free DNA Whole Exome Sequencing Data for Advanced Non-Small Cell Lung Cancer

**DOI:** 10.3390/ijms232112932

**Published:** 2022-10-26

**Authors:** Ho Jang, Chang-Min Choi, Seung-Hyeun Lee, Sungyong Lee, Mi-Kyung Jeong

**Affiliations:** 1Korea Medicine Data Division, Korea Institute of Oriental Medicine, Daejeon 34054, Korea; 2Department of Pulmonary and Critical Care Medicine, Department of Oncology, College of Medicine, University of Ulsan, Asan Medical Center, Seoul 05505, Korea; 3Division of Pulmonary and Critical Care Medicine, Department of Internal Medicine, Kyung Hee University Medical Center, Kyung Hee University School of Medicine, Seoul 02447, Korea; 4Division of Pulmonary, Allergy, and Critical Care Medicine, Department of Internal Medicine, Korea University Guro Hospital, Seoul 08308, Korea; 5Korea Medicine Convergence Research Division, Korea Institute of Oriental Medicine, Daejeon 34054, Korea

**Keywords:** plasma cell-free DNA, whole exome sequencing, copy number alteration detection, non-small cell lung cancer

## Abstract

Plasma cell-free DNA (cfDNA) sequencing data have been widely studied for early diagnosis and treatment response or recurrence monitoring of cancers because of the non-invasive benefits. In cancer studies, whole exome sequencing (WES) is mostly used for discovering single nucleotide variants (SNVs), but it also has the potential to detect copy number alterations (CNAs) that are mostly discovered by whole genome sequencing or microarray. In clinical settings where the quantity of the acquired blood from the patients is limited and where various sequencing experiments are not possible, providing various types of mutation information such as CNAs and SNVs using only WES will be helpful in the treatment decision. Here, we questioned whether the plasma cfDNA WES data for patients with advanced non-small cell lung cancer (NSCLC) could be exploited for CNA detection. When the read count (RC) signals of the WES data were investigated, a similar fluctuation pattern was observed among the signals of different samples, and it can be a major challenge hindering CNA detection. When these RC patterns among cfDNA were suppressed by the method we proposed, the cancerous CNAs were more distinguishable in some samples with higher cfDNA quantity. Although the potential to detect CNAs using the plasma cfDNA WES data for NSCLC patients was studied here, further studies with other cancer types, with more samples, and with more sophisticated techniques for bias correction are required to confirm our observation. In conclusion, the detection performance for cancerous CNAs can be improved by controlling RC bias, but it depends on the quantity of cfDNA in plasma.

## 1. Introduction

Plasma cell-free DNA (cfDNA) has been studied widely for the early diagnosis and response or recurrence monitoring of cancer. cfDNA is a liquid biopsy and has the potential to help identify cancer-related genetic variations such as copy number alterations (CNAs) and single nucleotide variants (SNVs) for the diagnosis and treatment of cancer patients from whom tissue biopsies are difficult to obtain [1]. The mutations from plasma cfDNA can be measured through droplet digital PCR (ddPCR) and next-generation sequencing (NGS) [2,3,4]. Although ddPCR can accurately detect mutations within a small amount of circulating tumor DNA (ctDNA), which is a type of cfDNA released from tumor cells, NGS technologies such as whole exome sequencing (WES) and whole genome sequencing (WGS) have advantages in terms of detecting more and diverse mutations because they cover wider genomic regions [1].

CNAs are duplications or deletions of genomic regions, and various somatic CNAs are observed in different cancer types [5]. Previous studies have detected CNAs in plasma cfDNA using low coverage WGS [6,7,8,9] and gene panel sequencing [10,11]. Although WES is mainly used for detecting SNVs which are the changes of a single nucleotide in cancer samples, the potential of CNA detection using WES in tumor tissues has been also addressed in several studies for cancer tissue sequencing [12,13,14].

In the clinical application of plasma cfDNA, the acquired blood quantity of the cancer patient is usually limited. In addition, various sequencing experiments are not possible due to limited DNA quantity in the blood sample. If not only SNVs but also CNAs can be detected in plasma cfDNA WES data, it will be beneficial for physicians and cancer patients.

Recently, cfDNA WES is used for investigating CNAs in a few studies in various cancers [15,16,17,18]. It implies that CNA information in cfDNA WES data can be a valuable option for cancer diagnosis and treatment. However, as far as we know, there is no study addressing the sequencing bias and detectability of CNAs in plasma cfDNA WES that might be closely related to the reliability of CNA analysis.

In this study, we studied sequencing bias in plasma cfDNA WES data and the detectability of CNAs in the data. Because CNAs are usually inferred from the number of mapped reads or read count (RC), bias in RC signals among multiple samples was investigated. We showed how much bias is shared among cancer samples, how this bias hinders CNA detection in cancer samples, and how CNA detection was improved by correcting bias. It is a novel study dealing with the challenges in CNA detection in plasma cfDNA WES data and giving the background for applying plasma cfDNA WES to clinical sequencing.

## 2. Materials and Methods

### 2.1. Cancer Patients and Plasma Samples

Plasma samples of advanced NSCLC patients who were enrolled for PD-1/PD-L1 inhibitor treatment were collected from 10 academic hospitals between August 2020 and March 2021. Baseline samples before undergoing treatment and samples during treatments were collected periodically.

### 2.2. Plasma cfDNA Extraction and WES

We collected 10 mL of blood in two 6 mL EDTA tubes. The samples were delivered within 24 h to the laboratory, and blood was separated into plasma and peripheral blood mononuclear cells. For each EDTA tube, 2 mL of plasma was separated by centrifugation at 1036× *g* for 5 min. For total 4 mL of plasma, 1.7 mL of plasma in upper part of the conical tube was separated by centrifugation at 2000× *g* for 5 to 10 min and stored in a cryotube at −80 °C. All DNA samples were quantified by PigoGreen (Invitrogen).

To generate a standard exome capture library, we used the Agilent SureSelect Target Enrichment protocol for the Illumina paired termination sequencing library. In all cases, the SureSelect Human All Exon V6 probe set was used. After the DNA was sequenced using the NovaSeq Platform (Illumina, San Diego, CA, USA) with 200× depth, FASTQ files were aligned to the human reference genome hg19 using BWA-MEM followed by mark duplicate step using Picard-tools and base quality score recalibration (BQSR) step.

### 2.3. RC Signal Quantification, Normalization, Denoising, and Segmentation

For quantifying RC, we set variable-size bins consisting of 100 uniquely mappable positions as in Wabico and BIC-seq2 [19,20]. We only considered uniquely mappable positions within exon regions because the WES data cover only exon regions rather than whole genomic regions in WGS. For each bin, the sum of the mapped reads on these positions will be the RC value of that bin. *RC*(*i*) indicates the number of mapped reads in bin *i*. We consider RC values in chromosomes 1 to 22 because the overall read counts of sex chromosomes are different between sexes. A total of 337,142 *RC* bins exist across 22 chromosomes.

To measure the level of CNAs, *RC*(*i*) was normalized to $NRC\left(i\right)=\frac{RC\left(i\right)}{Median(RC)}$, where Median(RC) is the median of all bins in the sample. NRC(i) close to 1 indicates neutral copy number, NRC(i) less than 1 implies copy number loss, and NRC(i) greater than 1 denotes amplified copy number.

To visually examine the pattern of NRC fluctuation and CNAs of the bins, the translation-invariant (TI) wavelet transform is applied to remove noise in the signal [21]. The decomposition level is 8, and the wavelet denoising parameter C is 2.0. See details of the wavelet denoising procedure in the study [19].

Circular binary segmentation (CBS) [22] was used to identify copy number altered genomic regions in NRC signals. R package DNAcopy [23] was used with default parameters.

### 2.4. RC Bias Correction by Suppressing Common Fluctuation Patterns

RC bias interferes with identifying sample specific CNAs, and thus, bias correction is necessary. The following equation shows bias-corrected and normalized read counts $RC_{corrected}\left(i\right)$ for a test sample.
$$RC_{corrected}\left(i\right)=Median\left(\frac{\frac{RC_{test}\left(i\right)}{Median(RC_{test})}}{\frac{RC_{sample,j}\left(i\right)}{Median(RC_{sample,j})}}\right)$$

$\frac{RC_{test}\left(i\right)}{Median(RC_{test})}$ is normalized read count of bin *i* for the test sample. $\frac{RC_{sample,j}\left(i\right)}{Median(RC_{sample,j})}$ is normalized read count of bin *i* for sample *j* that is used as control. Finally, the median across all corrections by each control sample is used for suppressing bias in bin *i*.

### 2.5. Cancerous CNA Detection Performance

For some samples, we measured how effectively cancerous CNAs are detected after bias correction in a manner similar to a previous study [24]. After suppressing common fluctuations in RC patterns in the test samples, we investigated the copy number of altered cancer genes already known to be related to lung cancer [24].

First, the genes of which $RC_{corrected}$ values are above the threshold value and that appear to be consistently detected in comparable independent samples in different WES sets were selected as silver standards. The threshold for amplified CNAs is as follows:$$Threshold_{amp}=Median\left(quant\left(0.9,\frac{\frac{RC_{sample,j}\left(i\right)}{Median(RC_{sample,j})}}{\frac{RC_{sample,k}\left(i\right)}{Median(RC_{sample,k})}}\right)\right)$$

For all samples *j* and *k* among the control samples in the WES set (*j* not equal to *k*), sample *j* is bias-corrected by sample *k*. The 90th quartile of the values of corrected bins will be the threshold for false amplifications for this case. The median of the quantile values for all possible *j* and *k* will be the threshold for these control sample sets. For the threshold for deleted CNAs ($Threshold_{del}$), the 10th quantile was used instead.

Next, we compare the detection performance before and after the bias correction by searching the silver standard genes. More specifically, after NRC and $RC_{corrected}$ signals of the test sample were segmented using CBS, we searched the silver standard genes in the segments in the order of neutral copy number segments to high-level copy number segments. The cumulative length of the CBS segments and the cumulative number of segments until the silver standard gene is found will be the cost for detecting the gene.

## 3. Results

### 3.1. Patient and Sample Information

Table 1 shows patient and sample information that was collected. The patients were being given PD-1/PD-L1 inhibitor medication. We collected 40 plasma samples from 16 patients with advanced NSCLC (stage 3 and 4). The DNA quantity ranged between 5 and 230 ng across all samples, and the median DNA quantity was 12.50. Figure 1 shows the distribution of DNA quantity for three independent sequencing experiments. Median DNA quantities for the three WES sets were 11, 14, and 10 ng, respectively. The DNA quantities of the samples from P13 patient for 2nd WES were far more than that of the other samples.

### 3.2. Similarities between RC Signals and Relation with cfDNA Quantity

Figure 2 shows the denoised RC signals for the samples in the 1st WES (for the signals in the 2nd and 3rd, see Appendix A). The signals are ordered according to the cfDNA quantities. Fluctuation patterns across the genome are similar among different samples. Table 2 shows correlations between all pairs of samples in the same WES set. Overall correlations in the 1st and 2nd experiments were lower than those in the 3rd experiment. Next, we compared the relation of signal correlations with cfDNA quantity.

Figure 3a,b show the relation of RC signal correlation and cfDNA quantities in the 1st WES set [25]. As shown in the figure, the samples whose cfDNA quantity was lower than others showed higher correlations. Some pairs with the samples of patients P11 and P13 showed lower correlations than the other pairs consistently. Moreover, their cfDNA quantities were the highest among all samples in the experiment (31 ng for P11 and 43 ng for P13). Similar patterns were observed in the 2nd experiment. The sample of these patients in the 2nd experiments also showed consistently lower correlations (Figure 3c). The cfDNA quantities of these samples also were highest in the 2nd experiment (20 ng for P11 and 230 ng for P13). However, in the 3rd WES set, although there was one sample whose cfDNA quantity was higher than 20 ng (P17-3), correlations with other samples were not low.

### 3.3. Cancerous CNA Detection by Suppressing Common Fluctuation

As described in the previous section, we observed the signal similarities among samples in the three WES sets by visualizing RC signals and calculating correlations between pair of signals. The signal bias represented in the form of similar fluctuations resembles true CNAs and hinders the detection of true CNAs that are important for the monitoring and treatment of cancers. Because the magnitude and patterns of fluctuations were similar for most samples in the experiments, we assumed that for some samples in the same WES experiment, the CNA detection performance can be improved by controlling the bias.

For the samples P11 and P13, which showed lower correlations with the other samples in the 1st and 2nd experiments, we generated $RC_{corrected}$ signals from $RC_{test}$ signals of these test samples. P01-1, P02-1, P03-1, P06-1, P07-1, P12-1, and P14-1 were used for control samples in the 1st WES set. P01-3, P02-3, P06-3, P07-3, P12-4, and P14-4 were used for control samples in the 2nd WES set. Sample P07-4 was not included because the same patient’s sample (P07-3) was already used as a control sample.

Figure 4a,b,e,f show the denoised bias-corrected signals ($RC_{corrected}$) of patients P11 and P13 (red color signals). The common fluctuations in the raw RC (black color) signals were suppressed, while unique CNAs of the patient were more distinguishable. For each patient, the corrected signals of two different WES sets are similar to each other ((a) and (b) for P11 and (e) and (f) for P13). All correlations between signals in the 3rd experiment are greater than 0.9. Although there is one sample whose cfDNA quantity is 26 ng (P17-3), the copy number levels in most genomic regions in denoised $RC_{corrected}$ are neutral across the genome (Appendix A).

In these patients, some alterations around lung cancer genes were consistently observed in two independent experiments. Figure 4c,d shows the RC signals on chromosome 3 of P11 sample in two independent experiments. The amplification around TERC genes were consistently observed in two WES sets. Figure 4g,h shows the amplification around chromosome 12 KRAS genes P13 samples in two independent experiments. For all cases, $RC_{corrected}$ (red color signal) became flatter than NRC (black color signal), implying a low number of false positives in CNA detection.

### 3.4. Improvement in CNA Detection

As described in the previous section, a similarity was observed between raw RC signals, and unique CNAs of the sample can be detected by controlling the sequencing bias. We measured how efficiently cancerous CNAs can be detected in the samples P11 and P13, as described in the Methods Section. Table 3 shows the silver standard genes manually curated in samples P11 and P13.

$Threshold_{amp}$ for the test samples in the 1st WES set was 1.271 and $Threshold_{del}$ was 0.786. $Threshold_{amp}$ for the 2nd WES set was 1.272 and $Threshold_{del}$ was 0.785. In the samples of P11, although genomic regions around ARNT, MCL1, and TERC were focally amplified consistently in two different WES sets, only CNA values of TERC were above the threshold. In case of P13, focal amplification around ARNT, MCL1, TERT, CCND3, VEGFA, and CCNE1 genes were observed in both WES experiments. However, the CNA values of CCND3 and VEGFA did not meet the threshold, and only five genes were considered as silver standards.

Figure 5 and Table 4 show the genomic length and the number of genomic segments required to detect silver standard genes in P13 samples. For both samples, silver standard genes were efficiently detected with smaller inspection length (Figure 5a) and fewer segments (Figure 5b) from bias-corrected RC signals (Blue for $RC_{corrected}$ of 1st WES and red for $RC_{corrected}$ of 2nd WES) than from raw corrected RC signals (black for NRC of 1st WES and yellow for NRC of 2nd WES). From the two samples, $RC_{corrected}$ signals can detect more silver standard genes with less inspection cost than NRC signals. Furthermore, the ARNT gene failed to be detected in the NRC of the 1st WES sample. In case of P11, since there existed only one silver standard gene (TERC), the inspection cost has not been provided in the figure.

## 4. Discussion

In this study, we investigated plasma cfDNA WES data bias and the detection of CNAs in three given WES datasets. For every WES set, we observed similar bias patterns in NRC signals in cfDNA samples, and the average correlations between pair of signals were more than 0.9 for each WES set, implying that there exists a bias in the RC signals of plasma cfDNA WES dataets. However, in the 1st and 2nd WES sets, the two samples P11 and P13, which had the highest amount of DNA quantities, showed the lowest correlations with the other samples in the same WES set. Although cancerous CNAs were observed in $RC_{corrected}$ signals of these samples, enough cfDNA quantity cannot always guarantee the detection of CNAs in cfDNA WES data. The cfDNA quantity of the sample P17-3 in the 3rd WES set was 26 ng, which is greater than that of the sample P11 in the 2nd WES set; however, distinguishable CNAs could not be observed in $RC_{corrected}$ signal in the sample P17-3. The detectability of CNAs may depend on various factors such as the patient’s health condition, type of cancer, status of cancer progression, proportion of ctDNA in cfDNA, and amount of collected plasma. To use the currently widely used NGS platform for detecting CNAs, the required cfDNA quantities need to be determined in future studies.

For controlling signal bias, we selected samples of patients with cancer in the same WES set in which cfDNA quantities were lower and signal correlation with other samples was higher. We assume that in the signal of these samples, level of CNAs are lower than the level of bias because of high correlation between these signals. However, there still exists a possibility of false positive CNA detection in the $RC_{corrected}$ signal due to true CNAs in the control samples. Hence, we set the conservative $Threshold_{amp}$ and $Threshold_{del}$ for determining the existence of CNAs in the test sample as described in the Methods Section. The question of whether false CNAs can be reduced if the samples of healthy individuals are provided instead of those of patients with cancer should be studied in the future.

Originally, the plasma cfDNA samples we used here were sequenced to investigate the possibility of monitoring changes of cancerous mutations such as SNVs and CNAs in cfDNA WES data for patients with advanced NSCLC who were enrolled for immunotherapies. The samples in the 1st WES set were baseline samples collected before the patient was given immunotherapy, and the samples in the 2nd WES set were collected after the first dose of immunotherapy to the same patients. Samples in the 3rd WES set included other patients’ samples before and after the immunotherapies. Except for two patients (P01 and P10), immunotherapies were stopped for others because they were diagnosed finally as PD or death during treatment. Because of the high similarities of WES data among samples as well as the sample imbalance (most patients ended as PD and only two patients were concluded as PR or SD), we cannot tell now whether mutation changes in patients with NSCLC can be monitored using cfDNA WES data.

As described in the Results Section, we mainly investigated signal similarities of samples within each WES experiment and bias corrections using the samples within the same WES set. Actually, fluctuation patterns looked similar across all samples in the three WES sets, as shown in Figure 2 and Appendix A. Hence, it is worth addressing whether the test samples can be bias-corrected by the control samples in the other WES experiment data as well as by the samples in the same WES set. Figure 6 shows the heat map and hierarchical clustering result of RC signal correlations across samples other than P11 and P13. Although the signal similarities were high across all samples (the minimum correlation was 0.9), the correlations among the samples in the same WES set seemed to be higher than that with the samples in the other sets. This may imply that the patterns of sequencing bias slightly vary in different WES experiments. A few more false-positive CNAs were observed when the test samples of the patients P11 and P13 were bias-corrected by the control samples in different WES sets than in the same WES set (data not shown).

In conclusion, plasma cfDNA WES data for patients with advanced NSCLC have the potential for detecting CNAs. A comprehensive analysis using more qualified samples is required to confirm the observations of this study and to develop a more sophisticated detection method in future.

## Figures and Tables

**Figure 1 ijms-23-12932-f001:**
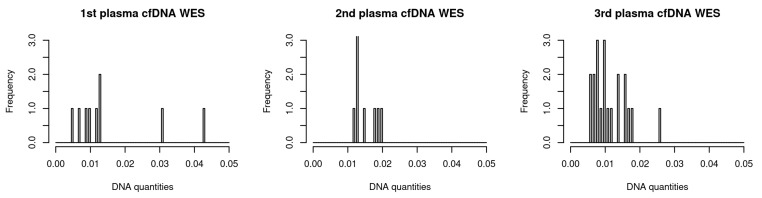
Histograms of DNA quantities in the three cfDNA WES sets. The DNA quantity of the sample P13-3 in the 2nd WES set was too high to be within the plot.

**Figure 2 ijms-23-12932-f002:**
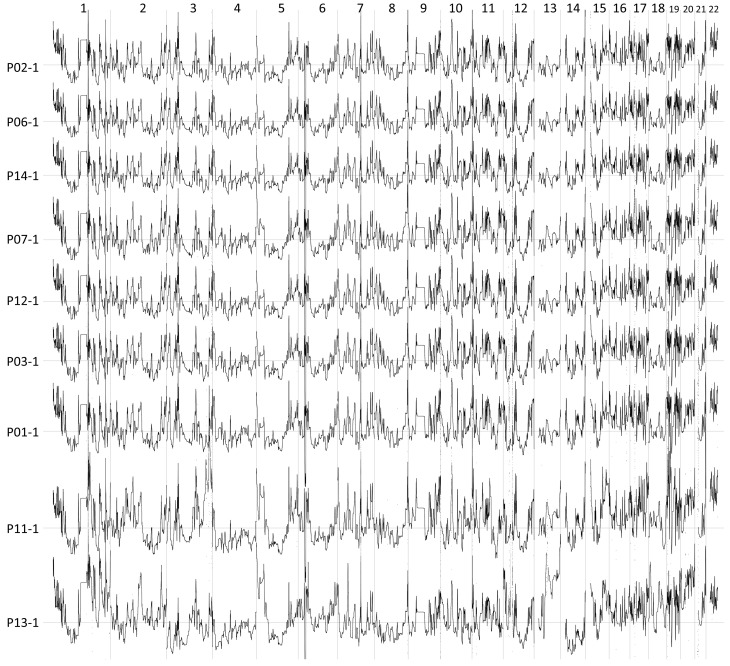
Read count patterns of the samples in the 1st cfDNA WES dataset. NRC signals were denoised using TI wavelet transformation and were visualized. The samples are sorted in order from the low (**Top**) to high (**Bottom**) DNA quantities.

**Figure 3 ijms-23-12932-f003:**
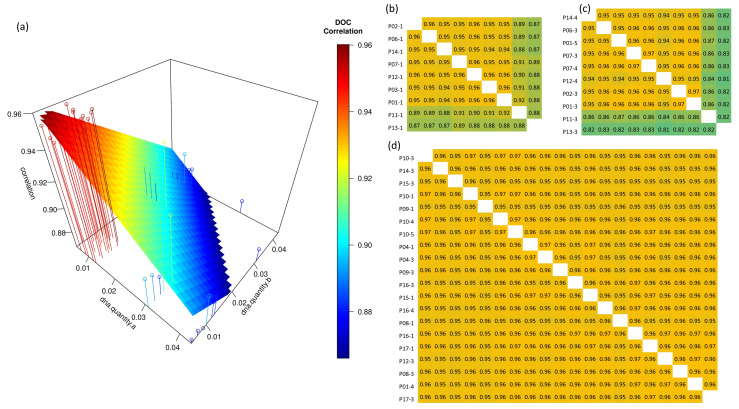
Relation of RC signal correlation and cfDNA quantities between all pairs of RC signals in three WES sets. (**a**) Three-dimensional (3D) plot showing the relations in the 1st cfDNA WES set. The x-axis indicates DNA quantity of one sample, the y-axis indicates DNA quantity of another sample, and the z-axis indicates the Pearson correlation between RC signals of these samples. (**b**–**d**) Heat map of correlations between signals in the 1st, 2nd, and 3rd WES sets. In the heat maps, samples were listed from low cfDNA quantity (**left**, **top**) to high cfDNA quantity (**right**, **bottom**). The closer to orange, the higher the correlation, and the closer to green, the lower the correlation. All heat maps are symmetric.

**Figure 4 ijms-23-12932-f004:**
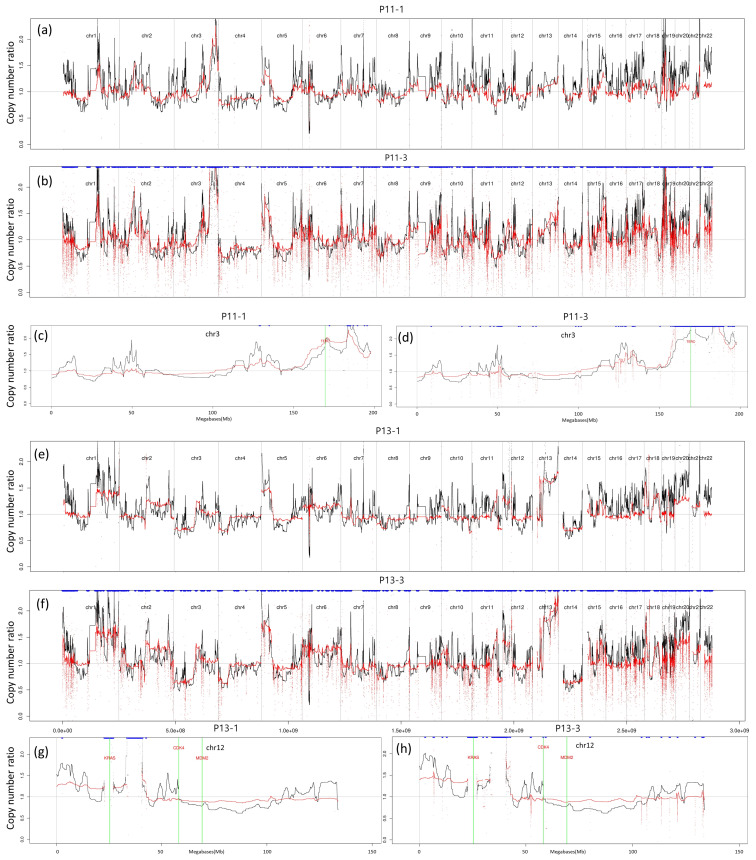
RC and $RC_{corrected}$ signals for P11 and P13 samples. (**a**) The figure shows RC (Black) and $RC_{corrected}$ (Red) signals for sample P11 in the 1st WES set. (**b**) The figure for P11 sample in the 2nd WES set. (**c**,**d**) The figures show the amplification around the TERC gene on chromosome 3 for sample P11 in the 1st and 2nd WES sets. (**e**) The figure for P13 sample in the 1st WES set. (**f**) The figure for P13 sample in the 2nd WES set. (**g**,**h**) The amplification around KRAS gene on chromosome 12 for sample P11 in the 1st and 2nd WES sets. The signals were denoised using TI wavelet transform.

**Figure 5 ijms-23-12932-f005:**
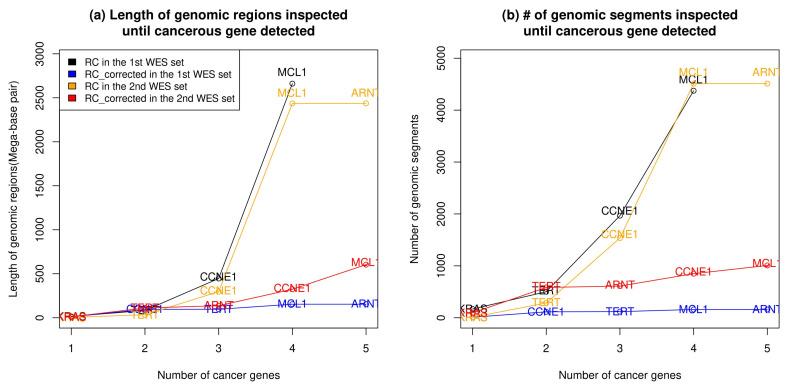
Total genomic length (**a**) and number of segments (**b**) for discovering cancerous genes from CBS segments of RC and $RC_{corrected}$ signal for the patient P13 in the 1st and 2nd WES sets.

**Figure 6 ijms-23-12932-f006:**
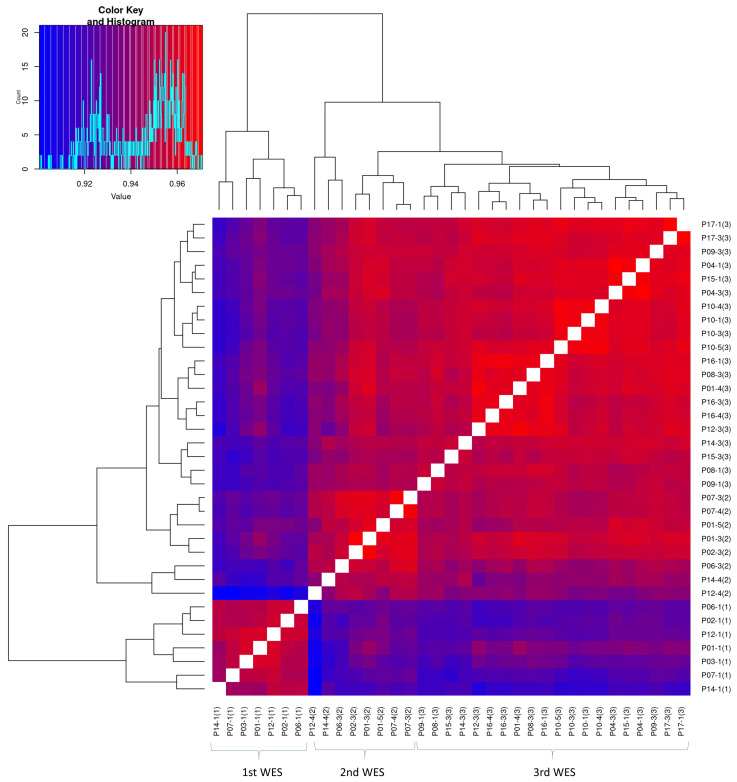
Heat map of RC signal correlation between samples in three WES sets except for P11 and P13 samples.

**Table 1 ijms-23-12932-t001:** Patient, WES sample, treatment, and clinical response.

Patient	1st WES	2nd WES	3rd WES	Immunotherapy	Clinical Response
P01	P01-1 (13 ng)	P01-3 (19 ng), P01-5 (13 ng)	P01-4 (18 ng)	Pembrolizumab	PR
P02	P02-1 (5 ng)	P02-3 (18 ng)		Atezolizumab	PD
P03	P03-1 (13 ng)			Atezolizumab	Death
P04			P04-1 (9 ng), P04-3 (10 ng)	Atezolizumab	PD
P06	P06-1 (7 ng)	P06-3 (13 ng)		Atezolizumab	PD
P07	P07-1 (10 ng)	P07-3 (13 ng), P07-4 (13 ng)		Pembrolizumab	PD
P08			P08-1 (14 ng), P08-3 (17 ng)	Atezolizumab	PD
P09			P09-1 (8 ng), P09-3 (10 ng)	Atezolizumab	PD
P10			P10-1 (7 ng), P10-3 (6 ng), P10-4 (8 ng), P10-5 (8 ng)	Atezolizumab	SD
P11	P11-1 (31 ng)	P11-3 (20 ng)		Pembrolizumab	PD
P12	P12-1 (12 ng)	P12-4 (15 ng)	P12-3 (16 ng)	Atezolizumab	PD
P13	P13-1 (43 ng)	P13-3 (230 ng)		Pembrolizumab	PD
P14	P14-1 (9 ng)	P14-4 (12 ng)	P14-3 (6 ng)	Pembrolizumab	PD
P15			P15-1 (11 ng), P15-3 (7 ng)	Atezolizumab	PD
P16			P16-1 (14 ng), P16-3 (20 ng), P16-4 (12 ng)	Atezolizumab	PD
P17			P17-1 (16 ng), P17-3 (26 ng)	Atezolizumab	Death

‘Patient’ denotes the patient codes. Note that P05 was not included in this study due to withdrawal of patient
consent. ‘1st WES’, ‘2nd WES’, and ‘3rd WES’ denote sample code with cfDNA quantity in three WES experiments,
e.g., ‘P01-3 (19 ng)’ in the 2nd WES means the plasma sample of the patient P01 collected from the 3rd visit and
the cfDNA quantity is 19 nanograms. ‘Immunotherapy’ is the type of immuotherapy that the patient received.
‘Clinical Response’ is the final response. PD, progressive disease; PR, partial response; SD, stable disease.

**Table 2 ijms-23-12932-t002:** Correlation between pair of samples within each WES set.

WES Exp	Num Pairs	Min.	Median	Mean	Max.	sd
1st WES	36	0.866	0.947	0.926	0.960	0.034
2nd WES	45	0.813	0.950	0.912	0.969	0.058
3rd WES	190	0.948	0.958	0.958	0.971	0.005

‘WES exp’ is the WES experiment. ‘Num Pairs’ is the total number of pairs of the samples in the set. ‘Min.’ is
minimum correlation value. ‘Median’ is median of correlation values. ‘Mean’ is mean correlation value. ‘Max.’ is
maximum correlation value. ‘sd’ is standard deviation of all correlations.

**Table 3 ijms-23-12932-t003:** CN-altered lung cancer genes in the patients P11 and P13. Those genes are consistently altered in corrected RC signals in 1st and 2nd WES independently.

Patient	Chr	Start	End	Name	Type	1st WES RC	2nd WES RC	Silver Standards
	1	150782181	150849244	ARNT	Amp	1.193	1.315	
P11	1	150547032	150552066	MCL1	Amp	1.193	1.643	
	3	169482308	169482848	TERC	Amp	1.831	2.287	Yes
	1	150782181	150849244	ARNT	Amp	1.423	1.645	Yes
	1	150547032	150552066	MCL1	Amp	1.423	1.348	Yes
	5	1253262	1295184	TERT	Amp	1.499	1.684	Yes
P13	6	41902671	42018095	CCND3	Amp	1.140	2.673	
	6	43737921	43754224	VEGFA	Amp	1.119	1.232	
	12	25357723	25403870	KRAS	Amp	5.281	7.571	Yes
	19	30302805	30315215	CCNE1	Amp	1.505	1.410	Yes

‘Patient’ denotes the patient codes. ‘Chr’, ‘Start’, and ‘End’ denote the chromosome, starting position and ending
position of a gene in the reference genome hg19. ‘Name’ is the gene name. ‘Type’ is the alteration type of the
gene. Note that all genes were CN-amplified. ‘1st WES RC’ and ‘2nd WES RC’ denote the *RC_corrected_* values of the
CBS segment overlapped with the gene. The values came from the samples in the 1st and 2nd WES sets. ‘Yes’ in
the ‘Silver Standard’ column means the gene is silver standard because the *RC_corrected_* values of the gene were
consistently above the *Threshold_amp_*.

**Table 4 ijms-23-12932-t004:** Total genomic length and number of segments for discovering cancerous genes for the patient P13 in the 1st and 2nd WES sets.

NRC (1st WES)			${\mathrm{RC}}_{\mathrm{corrected}}$ (1st WES)			NRC (2nd WES)			${\mathrm{RC}}_{\mathrm{corrected}}$ (2nd WES)		
**Gene**	**Length**	**Segments**	**Gene**	**Length**	**Segments**	**Gene**	**Length**	**Segments**	**Gene**	**Length**	**Segments**
KRAS	15.439	168	KRAS	10.810	10	KRAS	4.426	10	KRAS	10.928	94
TERT	82.933	502	CCNE1	95.528	114	TERT	34.160	285	TERT	110.974	584
CCNE1	445.820	1967	TERT	96.499	118	CCNE1	296.426	1536	ARNT	136.793	610
MCL1	2661.222	4374	ARNT	153.870	159	ARNT	2436.477	4511	CCNE1	323.650	849
			MCL1	153.870	159	MCL1	2436.477	4511	MCL1	603.770	1007

‘Gene’ denotes detected gene. ‘Length’ denotes the genomic length for detecting the gene. ‘Segments’ denotes the
number of CBS segments for detecting the gene.

## Data Availability

The data presented in this study are available on request from the corresponding author. The data are not publicly available due to containing information that could compromise the privacy of research participants.

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
