# Peer review of "Read Count Patterns and Detection of Cancerous Copy Number Alterations in Plasma Cell-Free DNA Whole Exome Sequencing Data for Advanced Non-Small Cell Lung Cancer"

_ijms, 2022, doi:10.3390/ijms232112932_

Round 1

Reviewer 1 Report

A well-designed article related to an issue of great interest. Although you tried to offer an encouraging conclusion,the method requires improvements in terms of the quantity required in a sample and the analysis of the obtained data, I would be happy if in the future you do an analysis of new data after you have corrected the shortcomings of the method. You will thus be able to achieve a better implementation of the method in practice.

Author Response

Thank you for your comments. We plan to develop more sophisticated methods with more qualified samples.  We added our future plan in the Section “3. Discussion” as follows.

Page 12 (Line 256)(3. Discussion): “A comprehensive analysis using more qualified samples is required to confirm the observations of this study and to develop a more sophisticated method in future.”

Reviewer 2 Report

The authors have presented a well written report on what are very difficult and technically demanding studies. Despite the in vitro and theoretical proposal that shed cfDNA from tumours are going to be invaluable  biomarkers; achieving reliable genetic marker data from real-life clinical samples is not straightforward.   The authors highlight that reliable results are going to be entirely dependant on the quality and quantity of cfDNA recovered AND the need to have correction features/filter to clean up the data when processed. I completely agree with that conclusion.

Two very very minor points: 

Line 66. Centrifugation should be reported in g or rcf (relative centrifugal force) not rpm ref is dependant on the rotor size. Decide on a time was it 3 or was it 5 mins. 

Line 69. The DNA quantification is inadequately described was this pico green satining of gels and crude estimation from the gel or a liquid fluorimeter picogreen standard curve generated and samples plotted against. (just refer to papers i.e quantification was my picogeen staining (ref) and quality of each sample dsDNA confirmed by gel electrophoresis (ref)).

Author Response

The authors have presented a well written report on what are very difficult and technically demanding studies. Despite the in vitro and theoretical proposal that shed cfDNA from tumours are going to be invaluable  biomarkers; achieving reliable genetic marker data from real-life clinical samples is not straightforward.   The authors highlight that reliable results are going to be entirely dependant on the quality and quantity of cfDNA recovered AND the need to have correction features/filter to clean up the data when processed. I completely agree with that conclusion.

Two very very minor points:

  1. Line 66. Centrifugation should be reported in g or rcf (relative centrifugal force) not rpm ref is dependant on the rotor size. Decide on a time was it 3 or was it 5 mins.

Answer) Thank you for your comment.  We modified the rpm value to the value converted to g as follows.

Page 2 (Line 66)( 1. Materials and Methods): “For each EDTA tube, 2 mL of plasma was separated by centrifugation at 1,036g for 5 min.”

  1. Line 69. The DNA quantification is inadequately described was this pico green satining of gels and crude estimation from the gel or a liquid fluorimeter picogreen standard curve generated and samples plotted against. (just refer to papers i.e quantification was my picogeen staining (ref) and quality of each sample dsDNA confirmed by gel electrophoresis (ref)).

Answer) Thank you for your comment. We clarified the meaning of the sentence by removing unnecessary words and adding the reference of the technique as you recommended.

Page 2 (Line 68)( 1. Materials and Methods): “All DNA samples were quantified by PigoGreen (Invitrogen).”